# The Therapeutic Potential of Supersulfides in Oxidative Stress-Related Diseases

**DOI:** 10.3390/biom15020172

**Published:** 2025-01-23

**Authors:** Yuexuan Pan, Tetsuro Matsunaga, Tianli Zhang, Takaaki Akaike

**Affiliations:** 1Department of Environmental Medicine and Molecular Toxicology, Tohoku University Graduate School of Medicine, Sendai 980-8575, Japan; pan.yuexuan.r6@dc.tohoku.ac.jp; 2Center for Integrated Control, Epidemiology and Molecular Pathophysiology of Infectious Diseases, Akita University, Akita 010-8543, Japan; matsunag@med.akita-u.ac.jp

**Keywords:** supersulfides, persulfides, polysulfides, reactive sulfur species, oxidative stress, reactive oxygen species, lipid peroxidation, oxidative stress-related diseases, systemic inflammatory response syndrome, chronic obstructive pulmonary disease

## Abstract

Oxidation-reduction (redox) reactions are fundamental to sustaining life, with reactive oxygen and nitrogen species playing pivotal roles in cellular signaling and homeostasis. However, excessive oxidative stress disrupts redox balance, contributing to a wide range of diseases, including inflammatory and pulmonary disorders, neurodegeneration, and cancer. Although numerous antioxidant therapies have been developed and tested for oxidative stress-related diseases, their clinical efficacy remains limited. Here, we introduce the emerging concept of ‘supersulfides’, a class of redox molecule species with unique antioxidant and nucleophilic properties, which have recently been recognized as crucial regulators of cellular redox homeostasis. Unlike traditional antioxidants, supersulfides offer novel mechanisms of action that directly target the underlying processes of oxidative stress. This review summarizes current knowledge on supersulfides, highlighting their roles in oxidative stress and associated diseases, as well as the mechanisms underlying oxidative stress-related pathology. The therapeutic potential of synthetic supersulfides for treating oxidative stress-related diseases is also discussed. A comprehensive understanding of the molecular and cellular basis of redox biology can help to guide the development of innovative redox-based therapeutic strategies aimed at preventing and treating diseases associated with disturbed redox regulation.

## 1. Introduction

Oxidation-reduction (redox) reactions are fundamental to the existence of life across species [1]. Various redox reactions regulate and drive essential processes, such as energy production, biosynthesis, and cellular signaling, through mechanisms collectively referred to as redox regulation [2,3,4,5]. These processes involve the transfer of electrons among reactive species, such as oxygen, nitrogen, sulfur, carbon, selenium, electrophiles, and halogens [6,7,8,9]. In these species, reactive oxygen species (ROS)–comprising radicals (e.g., superoxide (O_2_•^–^)) and non-radical derivatives (e.g., hydrogen peroxide (H_2_O_2_))–can engage in complex chemical interactions with other reactive species before interacting with larger biomolecules, such as nucleic acids, proteins, lipids, or carbohydrates [2,10,11] (Figure 1). One such example is the reaction between O_2_•^–^ and the free radical nitric oxide (•NO) generating the reactive nitrogen species (RNS) peroxynitrite (ONOO^–^), a chemical that has potentially harmful effects [12,13,14]. ROS and RNS contribute to oxidative eustress (beneficial stress) at low to intermediate concentrations, a process that is essential for maintaining cellular functions, adaptation, and resilience in response to physiological stimuli including exercise or changes in nutritional status [14,15]. At higher levels, excessive ROS production or an imbalance between oxidant generation and neutralization disrupts redox homeostasis, which can lead to oxidative distress (harmful stress), damaging biological systems and further contributing to diseases ranging from tissue-specific injury to systemic inflammation [14,16,17]. ROS and RNS signaling processes must therefore be maintained in order to avoid oxidative damage and this is regulated by an array of antioxidant systems within the cell [12,18,19].

At the onset of life on Earth approximately 3.8 billion years ago, the environment was devoid of oxygen. Sulfur is believed to have played a critical role in early respiration on Earth and it remains vital to biological systems, as it is incorporated into essential biomolecules, including amino acids, proteins, enzymes, and vitamins [20,21]. The versatile chemistry of sulfur enables compounds containing sulfur to act as effective antioxidants through the stabilization of unpaired electrons [22,23,24]. Advances in mass spectrometry-based metabolomics have stimulated interest in sulfur-containing biomolecules, particularly reactive sulfur species (RSS), leading to a deeper understanding of their biological significance [25,26,27,28,29]. However, researchers have gradually recognized several issues with the use of the abbreviation RSS. Specifically, there is an ongoing debate about the inclusion of hydrogen sulfide (H_2_S) and disulfides in this category, and the term can easily be confused with that for the perthiyl radical, RSS• [30].

To improve the classification of sulfur-containing molecules, we propose a term, “supersulfides”, to describe species with catenated sulfur moieties, including hydropersulfides (RSSH), hydropolysulfides (RSS_n_H, n > 1), polysulfides (RSS_n_R, n > 1), and inorganic persulfides and polysulfides [31] (Figure 2). This review summarizes current knowledge on the chemistry and biology of supersulfides, highlighting their roles in both oxidative eustress and distress. An overview of the mechanisms by which oxidative distress contributes to pathological conditions is also presented, along with a discussion of the emerging therapeutic potentials of natural and synthetic supersulfides in treating oxidative stress-related diseases. An improved understanding of the molecular and cellular basis of supersulfide-driven redox biology can help to guide the development of innovative therapeutic strategies to prevent and treat diseases associated with disturbed redox regulation.

## 2. Occurrence, Biosynthesis, and Metabolism of Supersulfdes in Organisms

### 2.1. Occurrence and Quantification

The accurate quantification of supersulfides in biological specimens is the fundamental prerequisite for exploring supersulfide-based redox biology. The approach employed by Akaike and colleagues utilized monobromobimane to alkylate oxidation-labile supersulfides, exploiting their nucleophilic and electrophilic properties [26]. However, due to the hydrolysis equilibrium inherent in polysulfide species, the use of strong electrophilic alkylating agents, including monobromobimane, led to the acceleration of polysulfide degradation through hydrolysis, thereby compromising the precision of quantification [32]. A more suitable method without this limitation was later developed using β-(4-hydroxyphenyl)ethyl iodoacetamide (HPE-IAM) as a supersulfide-trapping agent [27]. In this approach, biological samples are reacted with HPE-IAM to alkylate endogenous supersulfides, and known amounts of isotope-labeled standards are then spiked into the same samples. This is followed by quantification of HPE-IAM-derivatized supersulfide adducts using tandem mass spectrometry in multiple reaction monitoring mode. The peak area ratios between the samples and standards then reveal the concentrations of supersulfides in analytes [33] (Figure 2).

These methods have demonstrated the abundant and ubiquitous distribution of typical low-molecular-weight (LMW) supersulfides, including cysteine (CysSH)-based cysteine persulfide/polysulfide (CysSSH/CysSS_n_H, n > 1), glutathione persulfide/polysulfide (GSSH/GSS_n_H, n > 1), cysteine trisulfide (Cys-SSS-Cys) as well as glutathione trisulfide (GSSSG), in both prokaryotic and eukaryotic organisms, including yeast, bacteria, mammals and humans, at submillimolar to millimolar levels [32] (Figure 2). Inorganic supersulfides, such as hydrogen disulfide (H2S2), trisulfide (H2S3), and polymeric sulfur (Sn), have also been reported as potentially existing in living systems [34]. Furthermore, supersulfides have been identified in protein-bound forms through thiol linkages in CysSH residues [29,35,36].

### 2.2. Biosynthesis

The biosynthesis of supersulfides in all living organisms depends fundamentally on enzymatic mediation. Cystathionine β-synthase (CBS), cystathionine γ-lyase (CSE), and 3-mercaptopyruvate sulfurtransferase (MPST or 3-MST), were previously thought to be the key enzymes responsible for supersulfide generation through the trans-sulfuration pathway [37,38]; however, certain Gram-negative bacteria, such as the *Salmonella* species, lack CSE, implying the existence of alternative synthetic pathways for supersulfides [39]. A study on mice with a triple knockout of CBS, CSE, and 3-MST supported this notion, revealing that supersulfides are still produced in the absence of these enzymes, further indicating the presence of compensatory or alternative mechanisms [40].

One such mechanism involves cysteinyl-tRNA synthetase (CARS), an enzyme canonically known for catalyzing the synthesis of cysteinyl-tRNA [41]. Two isoforms of this enzyme exist in mammalian cells: cytosolic CARS1 and mitochondrial CARS2, both of which have been identified as novel cysteine persulfide synthase (CPERS), possessing conserved cysteine persulfide-synthesizing activity across species [27,42] (Figure 3). Studies with recombinant CARS have demonstrated that mouse CARS1, human CARS2, and *Escherichia coli* CARS can produce CysSSH in the presence of the substrate L-cysteine [27]. Furthermore, mass spectrometry-based analyses have shown a significant reduction in supersulfide levels in CARS2 knockout cells. A similar study revealed that heterozygous knockout mice for CARS2 exhibited significantly lower levels of LMW supersulfides compared to wild-type mice [27].

As mentioned above, CARS, like other aminoacyl-tRNA synthetases, is primarily involved in cysteinyl-tRNA production utilizing CysSH. Unexpectedly, CARS also conjugated CysSSH with tRNA in the production of persulfidated cysteinyl-tRNA, enabling the incorporation of CysSSH into nascent polypeptide chains during protein translation [27]. Persulfidated proteins have also been detected in both bacteria and mammalian cells, further corroborating this finding [36,43] (Figure 3).

These studies suggest that CARS serves as a principal supersulfide-synthesizing enzyme which is highly conserved in both prokaryotes and eukaryotes.

### 2.3. Catabolism

The plasticity of supersulfides gives rise to a complex range of metabolic fates, which remain an active area of investigation. LMW supersulfides are primarily synthesized in mitochondria through CARS2-mediated actions. These supersulfides are subsequently oxidized by sulfide:quinone oxidoreductase (SQR). The oxidized supersulfides participate in sulfur–oxygen hybrid respiration within the electron respiratory chain [27,30,44,45]. Meanwhile, excessive metabolites from supersulfides, such as H_2_S, are catalyzed by SQR to mitigate sulfide toxicity [46] (Figure 3). H_2_S oxidation in mitochondria also involves other sulfur oxidation enzymes, including rhodanese and persulfide dioxygenase (ETHE1), which convert H_2_S to thiosulfate and sulfite. Sulfite oxidase then oxidizes these intermediates to sulfate, with molecular oxygen as an electron acceptor [47] (Figure 3).

On a broader level, protein-bound supersulfides are reduced to regenerate protein thiols, a process that is primarily mediated by the depersulfidation capability of the thioredoxin (Trx) and thioredoxin reductase (TrxR) systems [47,48]. Additionally, the Trx-related protein of 14kDa (TRP14) contributes to depersulfidation, particularly when Trx is engaged heavily in the peroxiredoxin system. The same study also reported that glutathione reductase functions as a depersulfidase [48]. However, it remains unclear whether other members of the redoxin family such as peroxiredoxins and glutaredoxins, which are activated or deactivated through the oxidation and reduction in catalytic thiols, play a role in regulating protein-bound supersulfides.

The wide distribution and tight regulation of supersulfides in various forms provide further evidence of their crucial roles in biological systems.

## 3. Versatile Physiological Potentials of Supersulfides in Oxidative Distress

Oxidative distress contributes to cellular damage via two major mechanisms. The first involves the overproduction of ROS, particularly H_2_O_2_, which serves as a primary factor in both cellular toxicity and damage. The second mechanism is the oxidative modification of macromolecules such as membrane lipids, structural proteins, enzymes, and nucleic acids, causing the disruption of redox signaling pathways [16,49,50]. It should be noted that these mechanisms have been extensively reviewed over the past decades and are beyond the scope of this discussion; readers may refer to related reviews for additional details [1,11,18,51].

### 3.1. Supersulfides Regulate Oxidative Distress by Reacting with ROS

Intracellular ROS are major byproducts of the mitochondrial respiratory chain as a result of electron leakage [1,51,52]. Mitochondria are also a major source of supersulfides in mammalian cells, mediated primarily by CARS2, hinting at a potential interaction between ROS and supersulfides. Indeed, supersulfides exhibit superior nucleophilicity and reducing capacity when compared to their corresponding parental thiols [53,54,55]. For example, GSSH, but not its parental thiols GSH or H_2_S, reduces H_2_O_2_ efficiently and protects cells from cell death induced by H_2_O_2_ [26]. It was reported by Kunikata et al. that the exogenous addition of GSSSG increased intracellular GSSH levels significantly, thereby suppressing cell death caused by H_2_O_2_ exposure [56].

In addition to H_2_O_2_ scavenging, supersulfides can also react with free radical species. RSS_n_H exists in its deprotonated form as alkyl polysulfide anion (RSS_n_^–^) which readily reacts with radical species to form alkyl polysulfide radicals (RSS_n_•) [57]. This recombines to yield the polysulfide RSS_2n_SR, effectively neutralizing free radicals [58]. These polysulfides can then be reduced by cellular thiols, such as GSH, which regenerate active RSS_n_H and serve as an efficient radical trapping agent [59,60]. A typical model demonstrates the influence of the anti-ferroptotic defense system in mammalian cells, as described below in Section 5.1 [59] (Figure 4).

Another example of the regulation of oxidative distress by supersulfides is revealed in its direct reaction with endogenous electrophiles, including 8-nitroguanosine 3′,5′-cyclic monophosphate (8-nitro-cGMP), an important secondary messenger in redox signaling [26,61,62]. Sawa et al. demonstrated the conversion of 8-nitro-cGMP to 8-mercapt-cGMP, a biologically active metabolite in vivo, by supersulfides [62]. Other endogenous electrophiles, such as nitrated fatty acids, prostaglandin J2 and HNE, exhibit similar reactions, suggesting the key role that supersulfides play in detoxification of electrophiles [63,64,65].

The traditional dogma holds that H_2_S scavenges ONOO^–^/ONOOH and plays a critical role in vivo [66]; however, no enzyme or pathway that exclusively synthesizes H_2_S has been identified in mammalian cells. Rather, recent evidence suggests that H_2_S exists in dynamic equilibrium with supersulfides in biological systems, leading to challenges in isolating the pure effects of H_2_S. Furthermore, H_2_S predominantly exists in biological structures in its dissociated form as hydrogen sulfide anion (HS^–^), a molecule which is significantly less reactive than the supersulfide anion RSS_n_^−^ [67]. These results suggest that the role of H_2_S as a redox signaling molecule may have been overestimated. Both inorganic and organic hydropersulfides (HSSH and RSSH) are major antioxidants that are generated endogenously, highlighting their critical role in cellular defense mechanisms.

### 3.2. Supersulfides Regulate Oxidative Distress by Modulating Proteins

Cysteine thiols in proteins are the primary targets for modification by oxidants and electrophilic agents, allowing them to serve as redox switches to control the structure and/or function of redox-regulated proteins [68,69,70]. Advances in quantitative redox proteomic techniques, which are able to differentiate between various thiol modifications, in addition to improvements in the detection via mass spectrometry of low-abundance proteins, have recently stimulated significant research in protein supersulfidation [71,72].

ETHE1, a persulfide dioxygenase, metabolizes GSSH and GSS_n_H into glutathione with simultaneous oxygen consumption [73]. The Cys247 of ETHE1 reportedly plays a critical role in the formation of supersulfide-bound protein. Polysulfidation at this residue, or its mutation to serine, abolishes persulfide dioxygenase activity [43]. It is, therefore, suggested that ETHE1 is involved in redox regulation through supersulfidation of its cysteine residue. Another example of protein supersulfidation is alcohol dehydrogenase 5 (ADH5), a bifunctional enzyme with both S-nitrosoglutathione reductase activity and formaldehyde dehydrogenase activity [35]. In this case, mutation of Cys174 to serine significantly inhibits ADH5 supersulfidation and abolishes its S-nitrosoglutathione reductase activity, causing an enhancement in NO signaling. This mutation, however, has no effect on the formaldehyde dehydrogenase activity of ADH5 [35].

The effects of persulfidation on protein function are extremely diverse. It may activate or inactivate specific proteins, enabling their adaption under changing conditions [36,74]. During oxidative eustress, oxidation of cysteine thiols by H_2_O_2_ leads to protein S-sulfenylation; however, oxidation of sulfenates by H_2_O_2_ can produce sulfinates [75]. These products are reversible by sulfiredoxins, but further oxidation by H_2_O_2_ irreversibly yields sulfonates, which causes protein function loss, a process typically observed during oxidative distress. The presence of supersulfides enables protein persulfidation; this protects protein thiols from irreversible oxidation by forming perthiosulfenic, perthiosulfinic, and perthiosulfonic derivatives [76]. A more significant aspect of these processes is that the modified proteins can be reduced back to native thiols, restoring protein function [29] (Figure 5).

### 3.3. Anti-Inflammatory Effects of Supersulfides

Numerous studies have extended this research beyond the antioxidative properties of supersulfides to investigate their anti-inflammatory activity [54,77,78,79,80,81,82].

Through a large-scale genome-wide association study, a CARS2 single nucleotide polymorphism was associated with a low-activity phenotype and an increased risk of atherosclerosis. It was also shown that cellular deletion of CARS2 caused a significant increase in the production of inflammatory cytokines in macrophages in response to lipopolysaccharide (LPS), a component of Gram-negative bacteria [81]. Another study revealed, using CARS2 heterozygous knockout mice, that CARS2 plays an anti-inflammatory role under various inflammatory conditions [79]. As CARS2 plays a fundamental role in supersulfide synthesis, these results suggest that supersulfides act as negative regulators in inflammatory responses.

Macrophages, which lack the cystine transporter xCT showed significantly lower levels of supersulfides after LPS stimulation when compared to wild-type macrophages [80]. Under the same conditions, the unavailability of the cystine substrate for supersulfide generation promoted pro-inflammatory gene expression in macrophages. This indicates that supersulfides form a negative feedback loop in the inflammation process [80]. Moreover, it was also shown that under pathological conditions, the NACHT, LRR, and PYD domains-containing protein 3 (NLRP3) inflammasome, a key mediator of acute inflammation, triggers a cytokine storm [83]. Although ROS are believed to be necessary for NLRP3 inflammasome activation, the detailed molecular mechanism of its role remains unclear [84]. Zhang et al. demonstrated that intracellular GSH and its supersulfides, including GSSH and GSSSH, are excreted from macrophages, causing the generation of ROS and subsequent NLRP3 inflammasome activation [78]. The extracellular addition of GSH suppressed the efflux of GSH and its derived supersulfides, causing a significant suppression of NLRP3 inflammasome activation [78]. The anti-inflammatory effects of supersulfides have, therefore, been strongly highlighted by these findings (Figure 6).

## 4. Limitations of Current Antioxidant Therapeutic Agents

The overproduction of ROS can result in oxidative stress, causing damaging effects and contributing to the pathology of various diseases in humans [85,86,87,88,89,90]. Despite the fact that many synthetic compounds with antioxidant activity have been developed and applied as therapies for oxidative stress-related diseases, their effectiveness has been disappointing [16] (Figure 7). The reasons for this are discussed below.

### 4.1. ROS Play Important Physiological Roles

A significant challenge in the usage of antioxidants is that ROS plays complex biological roles, including essential physiological functions. Scavenging ROS indiscriminately can, in many cases, exacerbate the pathology (Figure 7). For example, in animals, a brief period of cardiac ischemia–reperfusion generates OH•, impairing contractile function during reoxygenation. Although scavenging ROS improves contractile function, it also prevents the adaptation of cardiac function to repeated brief periods of ischemia [91]. Patients with chronic inflammation suffering from severe infections often rely on ROS to help kill pathogens through inducing inflammation [92]; therefore, effective antioxidant therapies may give rise to side effects, including increased susceptibility to infections and disruption of ROS-dependent inflammatory signaling.

### 4.2. Oxidative Stress Is Not the Primary Mediator of Disease Pathology

While tissue injury often increases ROS production and contributes to further damage, the primary pathological driver may not be oxidative stress, as it occurs in many cases secondary to other initiating factors (Figure 7). Consequently, antioxidant therapies alone may be insufficient to achieve therapeutic efficacy. Instead, combining antioxidants with treatments targeting the primary pathological processes, such as inflammation, metabolic dysregulation, or cellular signaling abnormalities, may offer a more comprehensive approach to disease management. One such example is the intricate link between cytokines and oxidative distress and their influence on physiological and pathological processes [93]. Cytokine production is frequently upregulated during oxidative distress, causing a heightened inflammatory response. The release of pro-inflammatory cytokines such as TNF-α, IL-1β, and IL-6, drive a cytokine storm, and in such cases, antioxidant therapy may alleviate oxidative stress but also fail to address the underlying cytokine storm, limiting its therapeutic efficacy [94].

### 4.3. Other Limitations

There are still numerous other possible explanations for the ineffectiveness of antioxidant therapies, including: some antioxidants fail to reach the sites of oxidative damage, even if they enter the tissues or organs; they may not adequately reduce oxidative damage despite reaching the correct site [88,95]; paradoxically, excessively high trial doses can promote oxidative stress [96]; the use of single agents may interfere with the uptake or metabolism of other therapeutic agents [97,98]; finally, many antioxidants lack the sufficient reactive capability to neutralize ROS in a timely manner [99] (Figure 7).

## 5. Supersulfide Donors as Potential Therapeutic Agents for Oxidative Distress-Related Diseases

Both natural and synthetic supersulfide donors, including GSSSG, cysteine trisulfide (CysSSSCys), N-acetylcysteine (NAC) tetrasulfide (NAC-S2), thioglucose tetrasulfide, and inorganic polysulfide sodium salts such as disodium di-, tri-, and tetra-sulfide, can enter inside cells and organs to donate sulfur atoms to intracellular acceptors such as GSH, causing an increase in the supersulfide contents of the cell [100,101] (Figure 8). In this section, several supersulfide donors with therapeutic potential for oxidative stress-related diseases are introduced, along with an exploration of the mechanisms behind their effects.

### 5.1. Neurodegenerative Diseases

Oxidative stress plays a fundamental role in neurodegenerative diseases such as Alzheimer’s disease and Parkinson’s disease [102]. The initiating causes of these diseases involve complex genetic and environmental factors that vary depending on the condition; nevertheless, common signatures of these diseases include elevated iron levels and neuronal loss, suggesting that ferroptosis is a common neurodegenerative mechanism [103].

Ferroptosis is driven at the cellular level by the iron-catalyzed free radical oxidation of PUFAs in lipids [104]. This is an autocatalytic process in which lipid peroxidation is initiated in the presence of polyunsaturated fatty acids, oxygen, and free radicals. Sustained lipid peroxidation leads to the accumulation of harmful molecules such as lipid peroxides, aldehydes, and their conjugates, ultimately resulting in rupture of the plasma membrane [105,106]. Cells containing oxidizable lipids, therefore, require defense mechanisms to mitigate this process [106]. The enzymatic recycling of small-molecule antioxidants such as α-tocopherol, coenzyme Q10, vitamin K, tetrahydrobiopterin, cystine, and GSH facilitates the reduction in lipid peroxyl radicals to lipid alcohols [106,107,108,109].

RSS_n_H can act as pluripotent free radical scavengers, acting not only by propagating the lipid peroxyl radical (LOO•) but also by neutralizing the initiating radicals (X•). However, this molecule also reduces the radical forms of other antioxidants, such as tocopherol (Toc•), enhancing the recycling capability [110]. LOO• is generally scavenged via hydrogen atom transfer reactions, in which antioxidant molecules donate hydrogen atoms (H•), reducing LOO• to its inactive form as LOOH. During this process, the antioxidant itself becomes a radical (A•), which is less reactive than LOO•; this can be eliminated to generate non-radical products, and bring a halt to lipid peroxidation. RSS_n_H is a superior H• donor to RSH, as discussed above in Section 3.1 [110] (Figure 4).

The efficacy of supersulfide donors in inhibiting lipid oxidation was demonstrated in numerous independent studies. Kaneko et al. revealed the inhibition of lipid peroxidation in human plasma via donation from inorganic supersulfides [111]. A similar study employed synthetic organic supersulfide donors to inhibit lipid oxidation in artificial liposomes [112]. Barayue et al. reported that natural supersulfides are able to scavenge lipid radicals to protect cells from ferroptosis [59], highlighting their therapeutic potential for ferroptosis-related diseases. An in vivo study reported the prevention of neurodegeneration in the spinal cord by GSSSG and the recovery of mice from delayed paraplegia after spinal cord ischemia [113]. Additionally, sodium trisulfide, an inorganic polysulfide donor, has been demonstrated to protect midbrain dopaminergic neurons from 1-methyl-4-phenylpyridinium-induced degeneration by ameliorating oxidative stress [114].

These studies are particularly noteworthy given that ferroptosis has recently emerged as an attractive therapeutic target for certain cancers [115]. However, most cancers give rise to upregulated anti-ferroptotic defense systems in order to escape ferroptosis. A promising strategy for cancer therapy could, therefore, involve lowering CARS-mediated supersulfide generation. In addition, glutathione peroxidase 4, as a critical regulator of ferroptosis, facilitates the reduction in lipid hydroperoxides to lipid alcohols [116]. Although the association between supersulfides and GPX4 remains to be elucidated, exploring therapeutic strategies aimed at this enzyme could provide complementary approaches for overcoming anti-ferroptotic defenses in cancer therapy.

### 5.2. Systemic Inflammatory Response Syndrome (SIRS)

SIRS is a disorder caused by an exaggerated inflammatory response to infectious pathogens or non-infectious insults, affecting the entire body [117]. The release of oxidants and inflammatory cytokines is provoked by SIRS, and the condition can lead to reversible or irreversible end-organ dysfunction and even death. One such form of SIRS caused by infection is sepsis, which has common features with oxidative stress and inflammation with SIRS caused by non-infectious insults, although sepsis has been more frequently studied.

Toll-like receptor (TLR)-mediated nuclear factor-kappa B (NF-κB) signaling is a key regulatory mechanism in the pathogenesis of sepsis, as it is essential for inducing the expression of inflammatory cytokines and inducible NO synthase (iNOS)-mediated oxidative stress, respectively [118,119] (Figure 9).

The binding of a ligand to a TLR causes adaptor molecules–such as myeloid differentiation primary response 88 (MyD88), Toll/IL-1 receptor (TIR) domain-containing adaptor protein (TIRAP), TIR domain-containing adaptor-inducing interferon-β (TRIF), and TRIF-related adaptor molecule (TRAM)–to bind the TIR domain of the receptor. This interaction phosphorylizes inhibitor kappa B kinases, resulting in the proteasomal degradation of NF-κB inhibitor α (IκBα) [120]. NF-κB then translocate into the nucleus to induce the release of inflammatory cytokines, for example tumor necrosis factor (TNF)-α [121]. During LPS recognition by TLR4, TRIF-mediated signaling activates NF-κB and also triggers the activation of interferon (IFN) regulatory factor 3 (IRF3), which induces IFN-β expression [122]. IFN-β binds to the IFN-α/β receptor on the producing cell itself or on neighboring cells, which causes the amplification of iNOS expression and NO production through the signal transducer and activator of transcription (STAT)-mediated signaling [123] (Figure 9).

Administration of LPS in a mouse model of septic shock decreased the survival rate to 20%. However, when treated with the supersulfide donor NAC-S2, the survival rate increased to 90% and ameliorated inflammatory responses, such as reduced TNF-α production. By contrast, mice treated with sodium hydrogen sulfide exhibited no improvement in survival rate, suggesting a superior anti-inflammatory effect of NAC-S2 over that of H_2_S [77]. Cell-based analyses demonstrated the suppression of LPS-induced phosphorylation of both IκBα and NF-κB in macrophages treated with NAC-S2. No inhibition of the phosphorylation of p38 mitogen-activated protein (MAP) kinase, c-Jun N-terminal kinase (JNK), or extracellular signal-regulated kinase (ERK) was observed as a result of NAC-S2 treatment, suggesting that supersulfides specifically target NF-κB signaling [77]. Additionally, the expression of iNOS was also inhibited by NAC-S2, which abolishes both IFN-β production and the downstream phosphorylation of STAT-1 [77]. GSSSG, another endogenous supersulfie donor, was shown to suppress LPS-induced inflammatory profiling in both human and mouse epithelial cells [124].

Patients with severe sepsis have exhibited significantly elevated plasma levels of HNE, along with a decrease in levels of antioxidants such as vitamin C, vitamin E, and GSH [125,126,127,128]. Supersulfide donors may therefore exert their therapeutic effects primarily through antioxidative mechanisms rather than the anti-inflammatory effects described above.

### 5.3. Chronic Obstructive Pulmonary Disease (COPD)

COPD, a chronic lung disease, is a major global health epidemic and the fourth-leading cause of death worldwide, the pathogenesis of which has been linked to oxidative stress [129]. Increased levels of both oxidants and lipid peroxidation products, such as 8-isoprostane, have been detected consistently in the exhaled breath condensate of patients with COPD, along with significantly elevated levels of HNE and its adducts by at least 50% in the airway and alveolar epithelial cells, endothelial cells, and neutrophils of COPD patients [130,131]. Additionally, the urinary level of 8-OHdG, a marker of DNA oxidation, is much higher in patients with COPD [132]. Collectively, these findings reveal the role of oxidative stress in the pathogenesis of COPD, both in the lung and systemically in patients with the disease.

COPD patients experience a progressive limitation of airflow, arising from two primary pathological processes: remodeling and narrowing of the small airways, and destruction of the lung parenchyma, which results in the loss of alveolar attachments due to emphysema [133]. Chronic inflammation in the lung periphery is implicated in these pathological changes, with the intensity of inflammation being correlated with the progression of the disease [89]. The inflammatory responses underlying COPD pathogenesis are both initiated and exacerbated by persistent exposure to oxidants [134]. Studies have indicated that this inflammatory response involves both the innate and adaptive immune systems, characterized by an increase in the number of neutrophils in the airway lumen, along with heightened numbers of macrophages, T lymphocytes, and B lymphocytes in the lung tissue [135] (Figure 10). COPD progression is often not ameliorated by current therapies in many patients, given that they target either free radicals or inflammatory responses independently, as opposed to addressing both simultaneously [136].

Supersulfides have been shown to play an important role in COPD using CARS2 heterozygous knockout mice subjected to viral infection (e.g., SARS-CoV-2) or elastase treatment [79]. The subjects in both models exhibited an increased severity of disease phenotypes compared with wild-type mice. The viral infection model was characterized by a marked increase in 8-OHdG and pro-inflammatory cytokines in CARS2 heterozygous knockout mice compared with wild-type mice. This result implies that the loss of endogenous supersulfides is correlated with oxidative stress and inflammation under COPD conditions [79].

Pathogens such as influenza A virus and SARS-CoV-2 have been treated with GSSSG and inorganic supersulfide donors, in which the supersulfides were shown to block viral entry into cells by directly inactivating viral spike proteins. Viral replication was also inhibited by the presence of supersulfides within cells by targeting key viral proteases in SARS-CoV-2, including the papain-like protease and the 3CL or main protease. Clinically, a reduction in supersulfide levels was observed in lung cells and epithelial lining fluid from patients with COPD [137]. Despite the lack of direct data in this study, the protective effects observed have been attributed to the combined anti-inflammatory and antioxidative properties of supersulfides described above. Thus, the multifaceted mechanisms of supersulfides described herein underscore their protective functions against COPD.

## 6. Limitations and Future Perspectives of Supersulfide-Based Therapy

While supersulfides hold immense potential in the field of oxidative stress-related diseases, several challenges remain that limit their translational application [138,139].

First, the precise mechanisms governing supersulfide metabolism and their interactions with other redox molecules are not yet fully understood, warranting further mechanistic studies. Additionally, the stability, bioavailability, and pharmacokinetics of supersulfide donors require optimization to ensure therapeutic efficacy while minimizing off-target effects.

Another significant limitation lies in the lack of effective delivery systems that can target supersulfide donors to specific tissues or cellular compartments affected by oxidative stress. Innovative drug delivery technologies, such as nanocarrier systems, may provide a solution to this challenge by enhancing tissue specificity and reducing systemic toxicity. Furthermore, while preclinical studies have demonstrated promising results, comprehensive clinical trials are necessary to validate the safety and efficacy of supersulfide-based therapies in human diseases.

Future research should also focus on identifying disease-specific biomarkers that can aid in patient stratification and therapeutic monitoring, thereby enabling personalized medicine approaches. Another potential area of investigation is the role of Coenzyme A in supersulfide biology. coenzyme A, as a low-molecular-weight thiol, plays a critical role in cellular biosynthesis and energy production. While its involvement in supersulfide biosynthesis and function has been proposed, detailed mechanistic insights and experimental evidence are currently lacking. Future studies should explore this relationship, particularly in the context of redox regulation and metabolic pathways. By addressing these challenges, the therapeutic potential of supersulfides can be fully harnessed, paving the way for innovative treatments targeting oxidative stress-related diseases.

## 7. Concluding Remarks

In this review, the chemical and biological properties of supersulfides, which are closely related to almost all life-sustaining phenomena, have been outlined. Mounting evidence indicates a dynamic change in supersulfide levels during various oxidative stress-related diseases. This may be explained by the interaction of supersulfides with ROS, RNS or other oxidants, which are excessively produced under these conditions. This review has described thoroughly significant therapeutic potential of supersulfides in addressing both oxidative stress and inflammation. One example of a promising strategy involves the manipulation of supersulfide levels using supersulfide donors; alternatively, inhibiting CARS or other supersulfide-producing enzymes may enable the regulation of supersulfide bioavailability. Given the wide-ranging potential offered by these therapies, deeper analyses of the mechanisms of action, targets, pharmacokinetics, and the potential side effects of supersulfide donors in disease models are essential. Finally, in order to enhance the efficacy of therapeutic agents, it is crucially important to develop drug delivery systems that enable supersulfide donors to target affected areas.

## Figures and Tables

**Figure 1 biomolecules-15-00172-f001:**
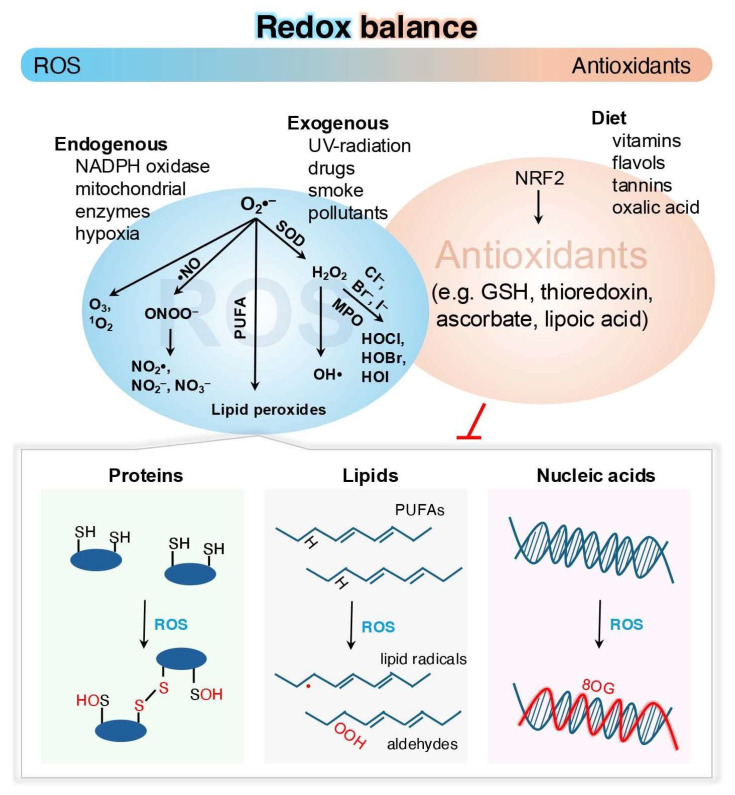
Redox balance and regulation. ROS are generated under the influence of external factors (e.g., UV radiation, pollutants) and internally by enzymes like nicotinamide adenine dinucleotide phosphate (NADPH) oxidases as well as mitochondrial respiration. ROS can modify cellular macromolecules such as proteins, lipids, or nucleic acids. Protein oxidation typically involves cysteine residues (P-SH) forming sulfenyl groups (P-SOH), which are groups that may react further to form disulfides or other modifications. Lipid peroxidation of poly-unsaturated fatty acids (PUFAs) generates lipid radicals and carcinogenic aldehydes like 4-hydroxy-2-nonenal (HNE). Oxidation of nucleic acid bases, particularly guanine to 8-hydroxy-20-deoxyguanosine (8OG), can lead to DNA damage. The body regulates ROS via enzymatic and non-enzymatic mechanisms, involving a diverse range of enzymes and LMW antioxidants (e.g., GSH and vitamins), which neutralize ROS and maintain redox homeostasis. Antioxidants are largely replenished through dietary intake. NRF2, nuclear factor erythroid 2-related factor 2; MPO, myeloperoxidase.

**Figure 2 biomolecules-15-00172-f002:**
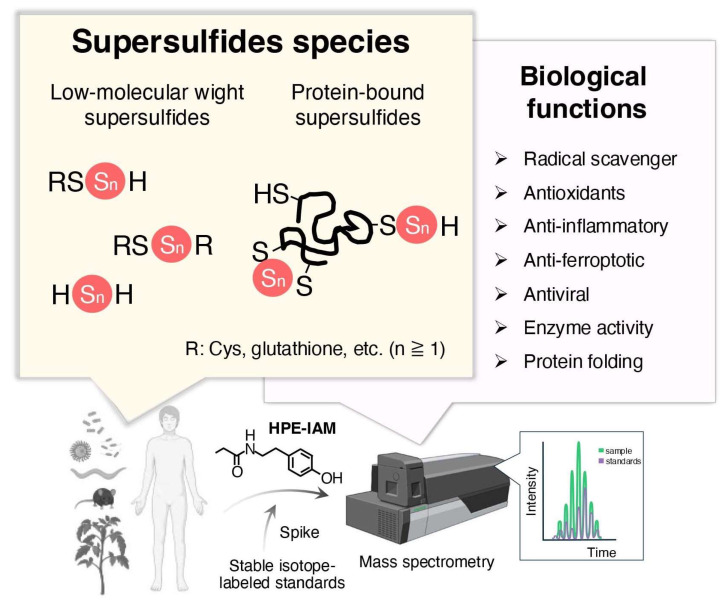
Occurrence and biological functions of supersulfides. Advances in mass spectrometry-based metabolomics have revealed the widespread presence of supersulfides in various organisms, existing as low-molecular-weight and protein-bound forms. Recent studies have uncovered diverse physiological functions of supersulfides, including their roles as radical scavengers, antioxidants, anti-inflammatory agents, anti-ferroptotic mediators, antiviral agents, and regulators of enzyme activity and protein folding.

**Figure 3 biomolecules-15-00172-f003:**
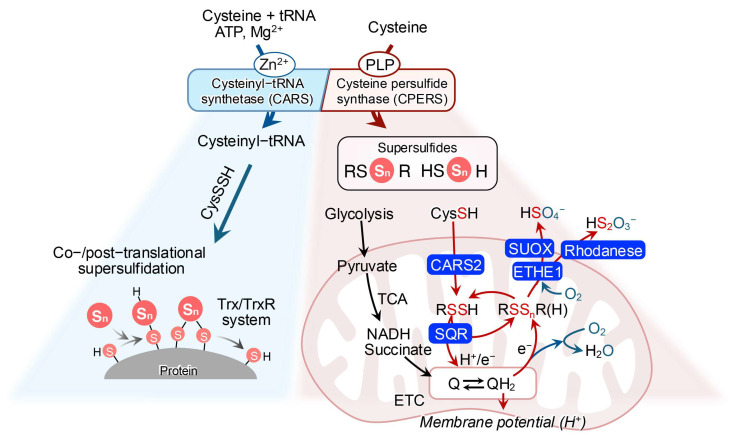
Dual functions of CARS. CARSs catalyze the production of CysSSH through a pyridoxal phosphate (PLP)-dependent reaction, utilizing a second cysteine molecule as the sulfur donor, independent of ATP and tRNA. This activity contributes to sulfur-oxygen hybrid respiration within mitochondria. Additionally, CARS also catalyzes the formation of tRNA-bound CysSSH adducts, enabling the incorporation of CysSSH into proteins. This process facilitates the generation of protein-bound supersulfides, a crucial step in translation-coupled protein supersulfidation. The dual functions of CARS are integral to cellular sulfur metabolism and redox regulation.

**Figure 4 biomolecules-15-00172-f004:**
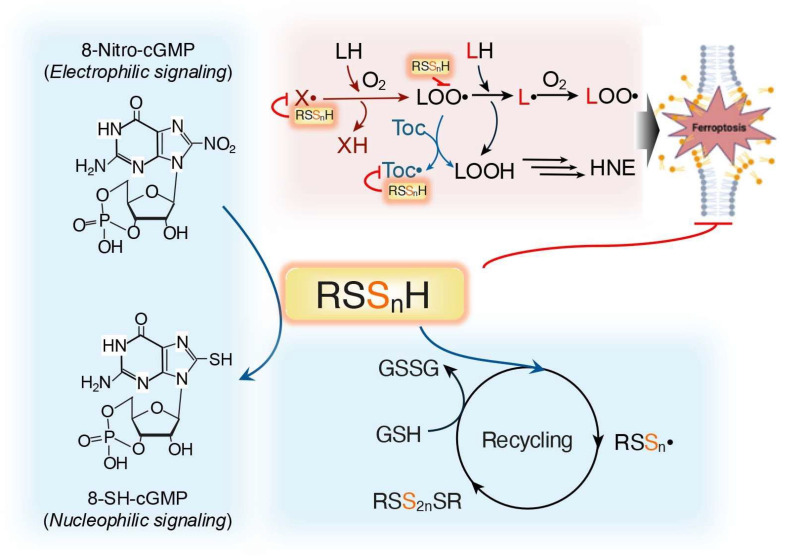
Supersulfides regulate oxidative and electrophilic stress. RSS_n_H prevent lipid peroxidation by scavenging the initiating radicals (X•) and propagating lipid peroxyl radicals LOO• or antioxidant radicals (e.g., Toc•), forming a key defense system against ferroptosis. Additionally, RSS_n_• intermediates are recycled by cellular GSH. Supersulfides also rapidly react with electrophilic chemicals, such as 8-nitro-cGMP, effectively mitigating electrophilic stress, as illustrated.

**Figure 5 biomolecules-15-00172-f005:**
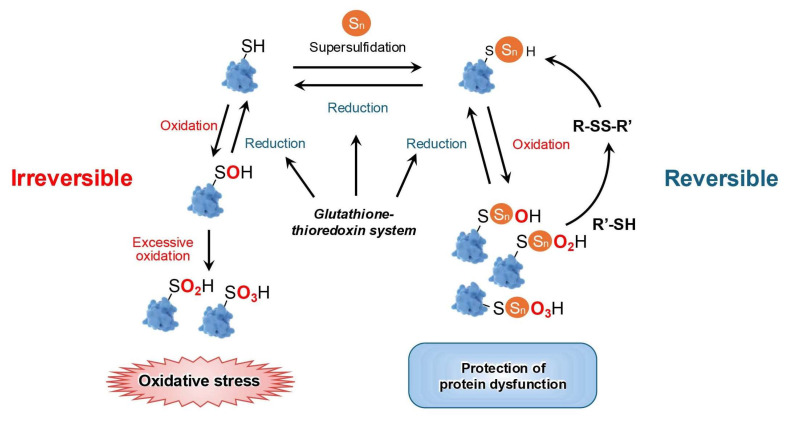
Restoration of protein function by supersulfides under oxidative stress. Protein thiols are initially oxidized to S-sulfenylated protein (Pro-SOH) under oxidative eustress. In the absence of regulatory mechanisms, these proteins can undergo irreversible overoxidation to form Pro-SO_n_H, leading to loss of function. However, supersulfides facilitating the formation of perthiosulfenic, perthiosulfinic, and perthiosulfonic proteins through supersulfidation, thereby reset the functions of proteins during excessive oxidative distress. The glutathione–thioredoxin system contributes to this process, ensuring redox homeostasis and protein functionality.

**Figure 6 biomolecules-15-00172-f006:**
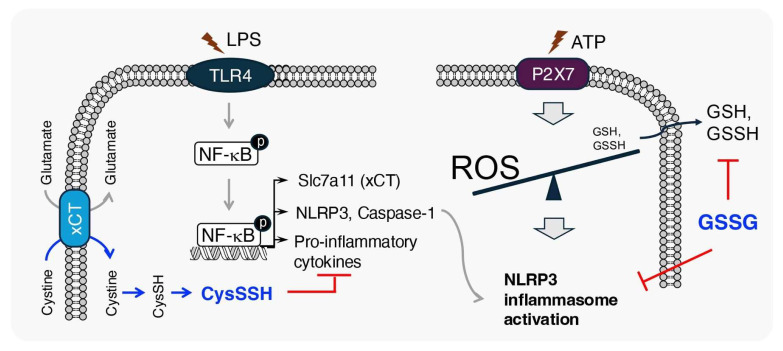
Endogenous supersulfides regulate inflammatory responses in macrophages. In activated macrophages, the expression of *slc7a11*, which encodes the cystine transporter xCT, is significantly upregulated upon LPS challenge. Increased xCT expression enhances cystine uptake, promoting the production of CysSSH and establishing a negative feedback loop to limit excessive inflammation. Concurrently, sensing danger-associated molecular patterns (e.g., ATP) induces the efflux of cellular GSSH, accompanied by ROS accumulation, leading to redox imbalance. This imbalance triggers NLRP3 inflammasome activation, amplifying inflammation. Notably, inhibition of GSSH efflux via exogenous GSSG administration significantly suppresses NLRP3 inflammasome activation, highlighting the regulatory role of supersulfides in inflammation. TLR4, Toll-like receptor 4; NF-κB, nuclear factor-kappa B; P2x7, P2X purinoceptor 7.

**Figure 7 biomolecules-15-00172-f007:**
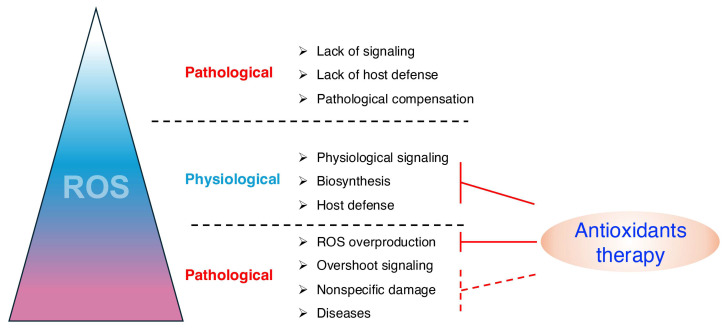
Limitation of traditional antioxidant therapy. ROS plays essential roles at moderate levels in maintaining physiological processes, including cell signaling, biosynthesis, and host defense mechanisms. Insufficient ROS levels can lead to impaired antimicrobial defense, hypotension, or loss of otoconia, whereas excessive ROS production is associated with pathological conditions such as cardiovascular disease, neurological disorders, cancers, and chronic inflammation. Traditional antioxidant therapies often fail to strike a balance, either excessively suppressing oxidative stress or neglecting to repair ROS-induced damage, thereby limiting their therapeutic efficacy.

**Figure 8 biomolecules-15-00172-f008:**
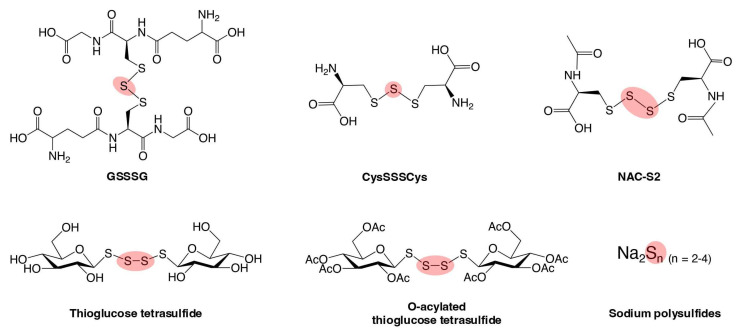
Synthetic supersulfide donors. The scheme illustrates various synthetic supersulfide donors, encompassing both organic and inorganic compounds. These chemical compounds serve as valuable tools for studying the biological roles of supersulfides and their therapeutic potential in pathological conditions.

**Figure 9 biomolecules-15-00172-f009:**
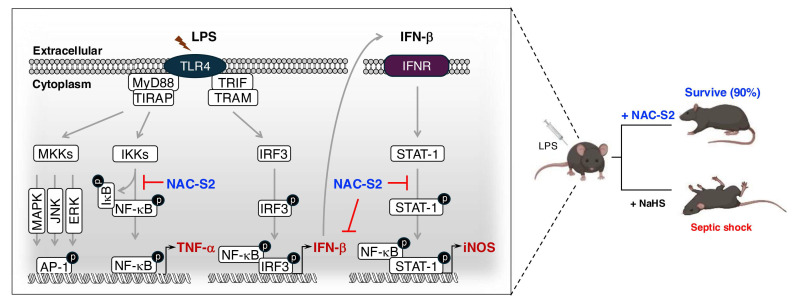
Therapeutic potential of supersulfides in sepsis. During LPS-induced sepsis, TLR4 signal transduction is initiated via the TIRAP-MyD88 and TRAM-TRIF pathways, leading to the expression of inflammatory mediators, including TNF-α, IFN-β, and iNOS, through the phosphorylation of NF-κB, IRF3, and STAT-1, respectively. Enhancement of cellular supersulfide levels by NAC-S2 potently inhibits the phosphorylation of these proteins, thereby suppressing the downstream inflammatory responses and protecting the mice from septic shock.

**Figure 10 biomolecules-15-00172-f010:**
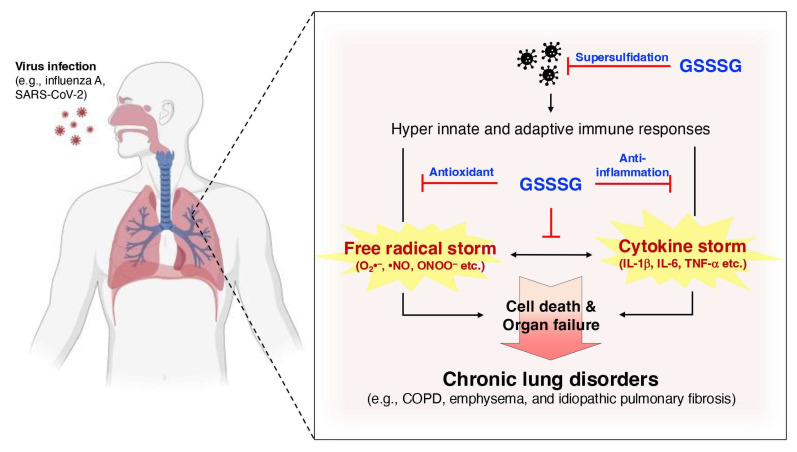
Protective roles of GSSSG in COPD. COPD, often triggered by viral infections such as influenza and SARS-CoV-2, is characterized by cytokine and radical storms in the lungs. The interplay between these storms exacerbates severe inflammation and oxidative distress. Endogenously derived or exogenously administrated GSSSG provides multi-faceted protection by inhibiting inflammatory responses, suppressing radical generation, and reducing viral infectivity and replication.

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
