# Peer review of "The Therapeutic Potential of Supersulfides in Oxidative Stress-Related Diseases"

_biomolecules, 2025, doi:10.3390/biom15020172_

Round 1

Reviewer 1 Report

Comments and Suggestions for Authors

This review summarises recent advances in the field of supersulfide biology and their potential for developing novel diagnostic and therapeutic approaches. The role of supersulfides in the regulation and maintaining redox homeostasis is highlighted, as compounds which can react with both electrophiles and nucleophiles. The biosynthesis of low molecular weight and protein-bound supersulfides is extensively discussed with the focus on the function of cytosolic and mitochondrial cysteinyl-tRNA synthases (CARS1/2). The review also covers the importance of supersulfides in scavenging ROS and protecting major cellular macromolecules (DNA, lipids and proteins) from irreversible oxidative damage, especially in pathologies associated with oxidative stress. 

The corresponding author of this manuscript is a world-leading expert in this emerging field of research, who coined the term of “supersulfides”. The review is well-written and logically structured and will attract the interest among your readerships. Therefore, this review deserves to be published in Biomolecules, when the authors address some critical comments and include changes in the manuscript.

Main comments and suggestions:

It is stated that supersulfides are highly abundant and methodologies have been developed to measure their cellular/tissue levels. The review would be strengthened if the quantification of the supersulfides in eukaryotic and prokaryotic cells is presented (possibly in a table).

It has been demonstrated that GSSG, and other LMW thiols in oxidised states, are produced in cellular response to oxidative stress. Furthermore, the GSH:GSSG ratio is a biological marker of oxidative stress. Is it the same for glutathione trisulfide or other supersulfides? If so, the expression or activation of CARS1/2 should be induced by cellular response to oxidative stress. 

CARS1/2 is involved in co-translational biosynthesis of supersulfide-modified proteins. How abundant are protein bound supersulfides? Are they predominantly translated under oxidative stress?

CARS1/2 are involved in the biosynthesis of CysSSH, but what is about GSSH, GSSG or GSSSG? 

Oxidative stress is not the main driver of disease pathogenesis. So, antioxidants can’t be used alone, but in combination with drugs targeting primary drivers of pathological processes.   

What is the percentage of co-translational supersulfidation and how it is controlled?

The GSH is the main antioxidant in mammalian cells and the GSH:GSSG has been considered as a marker of oxidative status. So, the review would benefit if the information is provided about the cellular ratio of GSH:GSSG:GSSSG under physiological conditions and oxidative stress. 

Coenzyme A (CoA) is critical for cellular biosynthetic processes and energy production. It functions as a substrate for diverse cellular enzymes and is required for utilising energy rich carboxylic acids (derived from the catabolism of nutrients) by generating metabolically active CoA thioesters. CoA is a LMW thiol and was recently shown to act as a major cellular antioxidant in all living cells. There is no doubt that it is involved in mediating supersulfide biology, including their biosynthesis and function. So, this topic should be discussed in the review. 

Vitamin K (known as ferroptosis suppressor protein 1, FSP1) – incorrect statement, update. 

The critical role of GPX4 in ferroptosis has to be also discussed in brief. 

The authors should also state whether AI-assisted technologies, such as ChatGPT-4, were used in the writing process to improve the readability and language of their own writing. 

Minor points: 

p16  - pathogenesis of numerous diseases, including … add neurodegeneration and cancer

p61 – The body regulates ROS via antioxidants – it would be more appropriate to state via enzymatic and non-enzymatic mechanisms, involving a diverse range of enzymes and LMW antioxidants.

p61  “can leads to DNA damage” – can lead to ….

p174  “of the redoxin”  - of the redoxin family

Author Response

This review summarises recent advances in the field of supersulfide biology and their potential for developing novel diagnostic and therapeutic approaches. The role of supersulfides in the regulation and maintaining redox homeostasis is highlighted, as compounds which can react with both electrophiles and nucleophiles. The biosynthesis of low molecular weight and protein-bound supersulfides is extensively discussed with the focus on the function of cytosolic and mitochondrial cysteinyl-tRNA synthases (CARS1/2). The review also covers the importance of supersulfides in scavenging ROS and protecting major cellular macromolecules (DNA, lipids and proteins) from irreversible oxidative damage, especially in pathologies associated with oxidative stress. 

The corresponding author of this manuscript is a world-leading expert in this emerging field of research, who coined the term of “supersulfides”. The review is well-written and logically structured and will attract the interest among your readerships. Therefore, this review deserves to be published in Biomolecules, when the authors address some critical comments and include changes in the manuscript.

Main comments and suggestions:

It is stated that supersulfides are highly abundant and methodologies have been developed to measure their cellular/tissue levels. The review would be strengthened if the quantification of the supersulfides in eukaryotic and prokaryotic cells is presented (possibly in a table).

Response: Thank you for your helpful suggestion. We agree with your point. Due to the variability of supersulfide levels across prokaryotic and eukaryotic organisms, as well as different cells and tissues, we have opted to revise the text to present the supersulfide concentrations as a range, rather than summarizing them in a table (line 110 in revised manuscript).

It has been demonstrated that GSSG, and other LMW thiols in oxidised states, are produced in cellular response to oxidative stress. Furthermore, the GSH:GSSG ratio is a biological marker of oxidative stress. Is it the same for glutathione trisulfide or other supersulfides? If so, the expression or activation of CARS1/2 should be induced by cellular response to oxidative stress. 

Response: As you pointed out, the GSH:GSSG ratio is a well-established marker of oxidative stress. In Ref 56, during influenza A virus infection in mice on the 7th day post-infection, the GSSG:GSH ratio was indeed elevated, indicating the occurrence of oxidative stress. However, in Supplementary Figure 3 of the same study, the GSSSG:GSH ratio appeared largely unchanged, while the GSSSG:GSSG ratio was slightly decreased during infection. In contrast, in CARS2 heterozygous knockout mice infected with influenza A virus, both the GSSSG:GSH and GSSG:GSH ratios increased over the course of the infection. These findings suggest that under certain specific conditions, the ratios of GSSSG, GSSG, and GSH may have potential as markers of oxidative stress. Additionally, as described in this manuscript and Ref 79, inorganic supersulfides may influence GSH biosynthesis in vivo, which could subsequently affect the ratios of GSSSG, GSSG, and GSH.

On the other hand, while CARS gene knockdown or knockout has been shown to affect LMW supersulfide levels, there is no direct evidence indicating that reduced endogenous GSSSG levels impact CARS expression. Therefore, we believe that further extensive research is required to clarify this question.

CARS1/2 is involved in co-translational biosynthesis of supersulfide-modified proteins. How abundant are protein bound supersulfides? Are they predominantly translated under oxidative stress?

Response: Thank you for this insightful comment. Protein-bound supersulfides play a crucial role in redox signaling and their levels vary according to the physiological state. Currently, Measuring the abundance of protein-bound supersulfides remains a challenge. A publication (PMID: 31914376) reported that oxidative stress conditions lead to increases significantly in protein persulfidation levels, suggesting that oxidative stress induces dynamic regulation of these modifications. Although direct data on CARS-mediated protein supersulfidation and oxidative stress are lacking, the involvement of CARS in this process may contribute to the formation of protein-bound supersulfides under oxidative conditions. Continued study is warranted to clarify it.

CARS1/2 are involved in the biosynthesis of CysSSH, but what is about GSSH, GSSG or GSSSG?

Response: As we mentioned in our response to your 2nd comment, there is currently no direct evidence identifying specific enzymes involved in the biosynthesis of GSSH or GSSSG. However, the concept of a “supersulfidome,” analogous to metabolomes and proteomes, may exist within cells and contribute to the diverse biological effects of supersulfides (Scheme in below). The Cys-SnH species are central to maintaining a dynamic supersulfidome, which encompasses reduced forms of supersulfides, including glutathione (GSnH, n > 1), proteins (Prot-SnH, n > 1), and inorganic polysulfides. We agree that is an important point and plan to further investigate this intriguing possibility in future studies.

Oxidative stress is not the main driver of disease pathogenesis. So, antioxidants can’t be used alone, but in combination with drugs targeting primary drivers of pathological processes. 

Response: We agree with this perspective and have revised Section 4.2 to highlight the necessity of combining antioxidants with therapies targeting primary pathological drivers (line 325 to 331 in revised manuscript). Thank you for your valuable suggestion.

What is the percentage of co-translational supersulfidation and how it is controlled?

Response: Thank you for raising this important point. Co-translational supersulfidation is indeed a fascinating area of study that remains underexplored. While precise quantification of co-translational supersulfidation is challenging, evidence from Ref 13 suggests that supersulfidation accounts for over 60% of modifications in proteins including GAPDH and ADH5.

Based on a publication (PMID: 31914376), enzymes such as cystathionine γ-lyase (CSE) and cystathionine β-synthase (CBS) play significant roles in intracellular persulfide generation, potentially influencing co-translational protein modifications. Additionally, Ref 15 in our manuscript highlights the critical role of reductive systems, including thioredoxin and glutathione, in maintaining persulfide homeostasis and preventing excessive accumulation.

These findings emphasize the involvement of enzymatic and redox mechanisms in regulating supersulfide dynamics. We recognize the importance of this question and plan to address it in future studies as detection technologies continue to advance.

The GSH is the main antioxidant in mammalian cells and the GSH:GSSG has been considered as a marker of oxidative status. So, the review would benefit if the information is provided about the cellular ratio of GSH:GSSG:GSSSG under physiological conditions and oxidative stress. 

Response: Thank you for raising this interesting and important point again. As we mentioned in our response to your 2nd comment, the cellular ratios of GSH and GSSG is indeed valuable markers for understanding redox dynamics. However, the precise quantification of GSH/GSSG/GSSSG ratios, particularly under varying physiological and oxidative stress conditions, remains a challenging area that requires further investigation. We greatly appreciate your suggestion and plan to explore this aspect in our future studies.

Coenzyme A (CoA) is critical for cellular biosynthetic processes and energy production. It functions as a substrate for diverse cellular enzymes and is required for utilising energy rich carboxylic acids (derived from the catabolism of nutrients) by generating metabolically active CoA thioesters. CoA is a LMW thiol and was recently shown to act as a major cellular antioxidant in all living cells. There is no doubt that it is involved in mediating supersulfide biology, including their biosynthesis and function. So, this topic should be discussed in the review. 

Response: Thank you for highlighting the importance of CoA. While we acknowledge its critical role in cellular biosynthesis and its potential relevance to supersulfide biology, current data of this topic is limited. Therefore, we have mentioned this point in section 6 at line 529 to 534, and we appreciate your understanding and will consider this important aspect in future studies.

Vitamin K (known as ferroptosis suppressor protein 1, FSP1) – incorrect statement, update. 

Response: Thank you for your comment. We have corrected the statement regarding Vitamin K and updated the sentence accordingly (Line 381 of revised manuscript).

The critical role of GPX4 in ferroptosis has to be also discussed in brief. 

Response: Thank you for your insightful comment. We completely agree with your perspective on the critical role of GPX4 in ferroptosis. In response to your suggestion, we have added a brief sentence discussing the importance of GPX4 in Line 407-411 of the revised manuscript and additionally cited a paper as Ref 81.

The authors should also state whether AI-assisted technologies, such as ChatGPT-4, were used in the writing process to improve the readability and language of their own writing.

Response: Thank you for your suggestion. We have included a statement in the Acknowledgments section confirming that AI-assisted technologies, such as ChatGPT-4, were not used in the writing process.

Minor points: 

p16  - pathogenesis of numerous diseases, including … add neurodegeneration and cancer

Response: Thank you for your suggestion. We have added (Line 14 of revised manuscript).

p61 – The body regulates ROS via antioxidants – it would be more appropriate to state via enzymatic and non-enzymatic mechanisms, involving a diverse range of enzymes and LMW antioxidants.

Response: We appreciate your valuable comments, and we have modified the sentences (Line 60-61 of revised manuscript).

p61  “can leads to DNA damage” – can lead to ….

Response: Thank you for your suggestion. We have made correction (Line 60 of revised manuscript).

p174  “of the redoxin”  - of the redoxin family

Response: Thank you for your suggestion. We have modified (Line 168 of revised manuscript).

Reviewer 2 Report

Comments and Suggestions for Authors

 Comments on the review paper entitled  “The therapeutic potential of supersulfides in oxidative stress-related diseases” submitted to Biomolecules

                 In this paper authors review the properties of sulfur-containing compounds and their ability to suppress oxidative stress. Authors propose introduce a new term “supersulfides” for persulfides and polysulfides instead widely used “reactive sulfur species”.  I agree that persulfides and polysulfides have properties a little distinct from H2S, however because of close relationship between H2S and per-, polysulfides, H2S could be also consider as reactive sulfur species. It can be concluded that the term “supersulfides” is very adequate for persulfides and polysulfides, while “reactive sulfur species” including H2S is “broader” concept.

This review highlights the role of supersulfides in oxidative stress and proposes potential of supersulfides in some oxidative stress-related diseases. The topic is important and relevant, however some points should be improved.

Below is a list of comments that the authors should follow to improve the manuscript.

1.       Figure 1 – Since authors write about 4-hydroxy-2-nonenal (HNE) I propose add “aldehydes” to lipid peroxides on the figure. The description of Figure 1 legend should be completed. Abbreviation PUFA should be introduced in the legend. Moreover, abbreviations NRF2 and MPO should be explained, and mentioned in the legend. The formulation “ROS are generated exogenously” is not exactly right, because in fact ROS are generated inside the cells under the influence exogenous factors. That's why I propose to replace it by the formulation “ROS are generated under the influence of external factors (UV radiation, pollutants)”

2.       Figure 2 – Since protein-bound supersulfides are presented separately, R should not be protein. Maybe “glutathione” would be better than “GS”? I propose to move the Figure 2 after line 119. Moreover, “quantification” should be added  to the title of subsection 2.1 (Occurrence and quantification)

3.       Figure 3 – the form of Protein-SSH should be presented on the figure or minus “-“should be added to “-S-Sn” to indicate protein persulfidation. Since Figure 3 is related also to persulfides catabolism it should be moved behind the section 2.3.

4.       Line 157 – Since biosynthesis of supersulfides was described in 2.2. subsection, the title of 2.3. subsection “Catabolism” would be better

5.       Figure 6 legend – it should be completed with description of symbols and abbreviations presented on the figure

6.       It would be better to place Figure 7 after discussion of this issue, that is behind subsection 4.3.

7.       In my opinion the subsection 5.1 should be supplemented with more data about the role of  supersulfides in neurodegenerative diseases. In fact, two first sentences treat about ferroptosis as a mechanism of neurodegenerative disorder and then only lines 392-393 are related directly to the role of supersulfides in neurodegeneration. The large part of this chapter is related to ferroptosis and RSSnH as free radical scavengers.

8.       Figure 9 should be more readable when it will be placed behind the chapter describing this issue.

9.       Figure 9 legend should be modified, because it presents mechanism of pathogenesis of sepsis and possible action of NAC-S2.

  Minor, technical points:

-          Line 33 – the space before [1] is needed

-          Line 122 – β instead b, γ- instead g

-          Line 179 – “supersulfides” instead “superuslfides”

-          Line 201 – “RSS2nSR” – 2 should be as subscript

-          Line 204 – “described” instead “descried”

-          Figure 4 – some elements (i.e. ferroptosis) is almost illegible

-          Line 217 – abbreviation HNE could be used

 -          Line 226 – “RSSn-“ is not supersulfide radical but supersulfide anion

 -          Figure 5 – The abbreviation Pro-SH does not exist on the figure, so it is unnecessary in the legend

 -          In some places NF-KB is used, while in another NF-kB

 -          Section 5.2 – IkB or IKBa? – it should be unified

 -          Line 434  - H2S could be used instead of the full name

 -          Line 436 – the appreciate reference would be better than citing Figure 9

 -          Lines 445-446 – the abbreviation HNE was introduced earlier

 -          Line 457 – the abbreviation 8OHdG was introduced earlier

 -          Line 460 – the citing the Figure 10 here in unnecessary

 -          Line 493 – “the papain-like protease and the 3CL or main protease” do not exist on the figure 10, so citing this figure here is unnecessary

Author Response

                 In this paper authors review the properties of sulfur-containing compounds and their ability to suppress oxidative stress. Authors propose introduce a new term “supersulfides” for persulfides and polysulfides instead widely used “reactive sulfur species”.  I agree that persulfides and polysulfides have properties a little distinct from H2S, however because of close relationship between H2S and per-, polysulfides, H2S could be also consider as reactive sulfur species. It can be concluded that the term “supersulfides” is very adequate for persulfides and polysulfides, while “reactive sulfur species” including H2S is “broader” concept.

This review highlights the role of supersulfides in oxidative stress and proposes potential of supersulfides in some oxidative stress-related diseases. The topic is important and relevant, however some points should be improved.

Below is a list of comments that the authors should follow to improve the manuscript.

  1. Figure 1 – Since authors write about 4-hydroxy-2-nonenal (HNE) I propose add “aldehydes” to lipid peroxides on the figure. The description of Figure 1 legend should be completed. Abbreviation PUFA should be introduced in the legend. Moreover, abbreviations NRF2 and MPO should be explained, and mentioned in the legend. The formulation “ROS are generated exogenously” is not exactly right, because in fact ROS are generated inside the cells under the influence exogenous factors. That's why I propose to replace it by the formulation “ROS are generated under the influence of external factors (UV radiation, pollutants)” 

Response: We appreciate your valuable suggestion. As your pointed out, we have added ‘PUFAs’, ‘lipid radicals’, and ‘aldehydes’ to Figure 1. Additionally, the abbreviations of PUFAs, NRF2, and MPO have been defined, and (P-SH) and (P-SOH) have been removed from the legend. The first sentence of the legend has also been revised for clarity.

  1. Figure 2 – Since protein-bound supersulfides are presented separately, R should not be protein. Maybe “glutathione” would be better than “GS”? I propose to move the Figure 2 after line 119. Moreover, “quantification” should be added  to the title of subsection 2.1 (Occurrence and quantification)

Response: Thank you for the critical comment. We have deleted ‘protein’ and replaced ‘GS’ with ‘glutathione’ in Figure 2. Additionally, this figure and its legend have been relocated after line 119 (line 115 to 121 in revised manuscript). The subtitle of Section 2.1 has also been modified to “Occurrence and quantification”.

  1. Figure 3 – the form of Protein-SSH should be presented on the figure or minus “-“should be added to “-S-Sn” to indicate protein persulfidation. Since Figure 3 is related also to persulfides catabolism it should be moved behind the section 2.3.

Response: Thank you for the suggestion. We have incorporated the form of ‘Protein-SSH’ into Figure 3, which has been relocated after Section 2.3 along with its legend (line 173 to 180 in revised manuscript).

  1. Line 157 – Since biosynthesis of supersulfides was described in 2.2. subsection, the title of 2.3. subsection “Catabolism” would be better

Response: Thank you for your suggestion. We have replaced ‘Catabolism’ with ‘Metabolism’.

  1. Figure 6 legend – it should be completed with description of symbols and abbreviations presented on the figure

Response: Thank you for the suggestion. We have descripted the abbreviations presented in Figure 6 (line 286 to 287 and 305 to 306 in revised manuscript).

  1. It would be better to place Figure 7 after discussion of this issue, that is behind subsection 4.3.

Response: Thank you for your kind suggestion. Figure 7 and its legend have been relocated after Section 4.3 (line 346 to 353 in revised manuscript).

  1. In my opinion the subsection 5.1 should be supplemented with more data about the role of  supersulfides in neurodegenerative diseases. In fact, two first sentences treat about ferroptosis as a mechanism of neurodegenerative disorder and then only lines 392-393 are related directly to the role of supersulfides in neurodegeneration. The large part of this chapter is related to ferroptosis and RSSnH as free radical scavengers.

Response: We are grateful to the reviewer for pointing this out. We have added one more evidence of protection of neurodegeneration by supersulfides in Line 400-402 and Ref79 of revised manuscript.

  1. Figure 9 should be more readable when it will be placed behind the chapter describing this issue. 

Response: Thank you for your kind suggestion. Figure 9 has been relocated (line 456-462 of revised manuscript).

  1. Figure 9 legend should be modified, because it presents mechanism of pathogenesis of sepsis and possible action of NAC-S2. 

Response: Thank you for your comment. We have rewritten the Figure 9 legend accordingly.

  Minor, technical points:

-          Line 33 – the space before [1] is needed

Response: Thank you for your careful review. We have inserted the space (Line 34 of revised manuscript).

-          Line 122 – β instead b, γ- instead g

Response: We apologize for these oversights and have made the corrections (Line 124 of revised manuscript).

-          Line 179 – “supersulfides” instead “superuslfides”

Response: We apologize for the mistake and have made the modification (Line 181 of revised manuscript).

-          Line 201 – “RSS2nSR” – 2 should be as subscript

Response: Thank you for your careful review. It has been modified (Line 203 of revised manuscript).

-          Line 204 – “described” instead “descried”

 Response: We apologize for the oversight and have made the corrections (Line 206 of revised manuscript).

-          Figure 4 – some elements (i.e. ferroptosis) is almost illegible

Response: We apologize for the inconvenience. All figures have been replaced with a clearer version for better readability.

-          Line 217 – abbreviation HNE could be used

Response: Thank you for your pointed it out. It has been modified (Line 219 of revised manuscript).

 -          Line 226 – “RSSn-“ is not supersulfide radical but supersulfide anion

Response: Thank you for your pointed it out. It has been modified (Line 228 of revised manuscript).

 -          Figure 5 – The abbreviation Pro-SH does not exist on the figure, so it is unnecessary in the legend

Response: Thank you for your suggestion. We have removed it.

 -          In some places NF-KB is used, while in another NF-kB

Response: Thank you for your suggestion. We have modified them.

 -          Section 5.2 – IkB or IKBa? – it should be unified

Response: As the suggestion, we have unified them.

 -          Line 434  - H2S could be used instead of the full name

Response: Thank you for the comment. However, sodium hydrogen sulfide is used to indicate a H2S donor; therefore, we have decided not to make any changes (Line 440 of revised manuscript).

 -          Line 436 – the appreciate reference would be better than citing Figure 9

Response: Thank you for your suggestion. We have removed ‘Figure 9’ from the sentence and added a reference (Line 442 of revised manuscript).

 -          Lines 445-446 – the abbreviation HNE was introduced earlier

Response: Thank you for your suggestion. We have replaced it with HNE (Line 452 of revised manuscript).

 -          Line 457 – the abbreviation 8OHdG was introduced earlier

Response: Thank you for your suggestion. We have replaced it with 8OHdG (Line 470 of revised manuscript).

 -          Line 460 – the citing the Figure 10 here in unnecessary

Response: As you pointed it out, we have removed it.

 -          Line 493 – “the papain-like protease and the 3CL or main protease” do not exist on the figure 10, so citing this figure here is unnecessary

Response: Thank you for your suggestion. We have removed it.

Reviewer 3 Report

Comments and Suggestions for Authors

The manuscript entitled "The therapeutic potential of supersulfides in oxidative stress-related diseases" authored by Yuexuan Pan et al. presents a very interesting and documented material on the therapeutic value found for the group of supersulfides for the selected case of oxidative stress -related diseases

The material is very well documented, as it is based on an extended literature research that also includes recent studies

The content of this review matches very well the aim, the authors present in a logical and scientific manner the importance of redox reactions and reactive oxygen species, followed by a structured presentation of sulfur-containing molecules (occurrence, biosynthesis, and metabolism of supersulfdes,  physiological potentials  in oxidative distress with associated mechanisms) 

Minor aspects to be corrected:

1. the abstract content is not very well balanced - please resume the initial part presenting the oxidative stress related (lines 11-17) and add the study novelty, as well as the conclusions of the review

2. The authors use figures, still tables summarizing the mechanism/diseases and the related studies (section 3 and section 5, respectively) would be recommended 

3. Limitations of supersulfide and also of studies on supersulfide, as well as future prospects should be added before the concluding remarks

Author Response

The manuscript entitled "The therapeutic potential of supersulfides in oxidative stress-related diseases" authored by Yuexuan Pan et al. presents a very interesting and documented material on the therapeutic value found for the group of supersulfides for the selected case of oxidative stress -related diseases

The material is very well documented, as it is based on an extended literature research that also includes recent studies

The content of this review matches very well the aim, the authors present in a logical and scientific manner the importance of redox reactions and reactive oxygen species, followed by a structured presentation of sulfur-containing molecules (occurrence, biosynthesis, and metabolism of supersulfdes,  physiological potentials  in oxidative distress with associated mechanisms) 

Minor aspects to be corrected:

1. the abstract content is not very well balanced - please resume the initial part presenting the oxidative stress related (lines 11-17) and add the study novelty, as well as the conclusions of the review

Response: We appreciate your valuable comments, and we have modified the sentence in the Abstract section (lines 11-20).

2. The authors use figures, still tables summarizing the mechanism/diseases and the related studies (section 3 and section 5, respectively) would be recommended 

Response: Thank you for your valuable suggestion. We believe the current figures and detailed textual descriptions already convey the key information effectively. Adding tables may inadvertently lead to redundancy with the existing figures. To maintain the flow and avoid excessive overlap, we propose retaining the current format while ensuring clarity in the text. We hope this approach is acceptable and welcome further suggestions if clarification is needed.

3. Limitations of supersulfide and also of studies on supersulfide, as well as future prospects should be added before the concluding remarks

Response: We are grateful to the reviewer for pointing this out. We have added section of limitations and future perspectives of supersulfide-based therapy in lines 512-536 of revised manuscript.
